# Multifunctional Magnetic Oxide Nanoparticle (MNP) Core-Shell: Review of Synthesis, Structural Studies and Application for Wastewater Treatment

**DOI:** 10.3390/molecules25184110

**Published:** 2020-09-09

**Authors:** Ebenezer C. Nnadozie, Peter A. Ajibade

**Affiliations:** School of Chemistry and Physics, University of Kwazulu-Natal, Private Bag X01, Scottsville Pietermaritzburg 3209, South Africa; 217081040@stuukznac.onmicrosoft.com

**Keywords:** magnetic oxides, nanocomposites, adsorption, wastewater, isotherms

## Abstract

The demand for water is predicted to increase significantly over the coming decades; thus, there is a need to develop an inclusive wastewater decontaminator for the effective management and conservation of water. Magnetic oxide nanocomposites have great potentials as global and novel remediators for wastewater treatment, with robust environmental and economic gains. Environment-responsive nanocomposites would offer wide flexibility to harvest and utilize massive untapped natural energy sources to drive a green economy in tandem with the United Nations Sustainable Development Goals. Recent attempts to engineer smart magnetic oxide nanocomposites for wastewater treatment has been reported by several researchers. However, the magnetic properties of superparamagnetic nanocomposite materials and their adsorption properties nexus as fundamental to the design of recyclable nanomaterials are desirable for industrial application. The potentials of facile magnetic recovery, ease of functionalization, reusability, solar responsiveness, biocompatibility and ergonomic design promote the application of magnetic oxide nanocomposites in wastewater treatment. The review makes a holistic attempt to explore magnetic oxide nanocomposites for wastewater treatment; futuristic smart magnetic oxides as an elixir to global water scarcity is expounded. Desirable adsorption parameters and properties of magnetic oxides nanocomposites are explored while considering their fate in biological and environmental media.

## 1. Introduction

The discovery of nanomaterials can be traced to the discussions of Richard Feynman in his 1959 historic seminar lecture entitled “there’s plenty of room at the bottom” [1]. The legendary submission of Feynman has since led to unprecedented discoveries in the field of science and technology with enormous possibilities in various fields of knowledge [2]. Recently, there has been heightened interest in magnetic nanomaterials because of their inherent unique properties. Magnetic nanocomposites belong to the group of nanomaterials that can be controlled with the application of a magnetic field. In wastewater treatment, spent magnetic nanocomposites can be effectively recovered from the treatment plant via magnetization by an external magnetic source for treatment and re-use. They contain a primary magnetic material and secondary component(s) that give functionality, normally called shell(s). Superparamagnetic nanocomposites are built with magnetic elements such as nickel, iron, cobalt or their oxides and alloys with ferromagnetic or ferrimagnetic characteristics as a core and an outer shell [3]. The coating and modification of magnetic nanoparticles helps to improve and protect the core against oxidation and instability while providing selectivity for the magnetic nanocomposite [4,5,6]. The shell components can be organic, inorganic or a blend of both. Silica, resin, alumina, polymers, polyelectrolytes, surfactants, carbon and hemoglobin have been utilized by various authors to modify superparamagnetic nanoparticles [7,8,9,10,11,12,13]. Magnetic nanocomposites have been adapted as a green means of transport in biological and environmental media based on ease of operation and overall economic gains [13]. The preparation methods of magnetic nanomaterials are instrumental to their inherent properties. Synthetic methods and characterization procedures have been extensively covered in a similar submission and will not be emphasized in this discussion [2]. The morphology of magnetic nanocomposites, such as shape, surface area, pore volume, mesoporosity and point of zero charge (PZC), are important considerations for an effective and efficient adsorption process. Organic nanocomposites are desirable because of their relatively large surface area and abundant functional active groups desirable for adsorption, while inorganic nanocomposites have narrow band gaps, wide band edges and high electron flow and conductivity [14,15]. The low removal rate of biochar for heavy metals has also been reported [16]. Magnetic nanocomposites would harness the properties of both organic and inorganic composites, making room for the sorption of a wide range of pollutants. Furthermore, the uniqueness of magnetic nanocomposites promotes the easy recovery and re-use of the adsorbent, colloid stability and environment compatibility.

### 1.1. Magnetic and Magnetic Oxide Nanocomposites

Engineered magnetic nanocomposites work on the principle of magnetic separation. The magnetic composite or fluid becomes magnetized in the presence of an external magnetic field and acts as a default in its absence. As the ferromagnetic particle size shrinks from multi-domain to a single domain, the thermal energy exceeds the energy barrier, which is responsible for the alignment of the magnetic material, resulting in superparamagnetism [17]. Superparamagnetism is dependent on the particle volume and anisotropy constant, for example, the single domain size for magnetite is 128 nm while the critical value for superparamagnetism is 10–20 nm [17,18]. This is a foundational principle for superparamagnetic nanocomposites. The coercivity of a magnetic material is related to the width of the magnetization curve; it tells us about the magnetic field strength that must be applied to bring the magnetization of the material back to zero. Remanence is the value of the magnetization of the magnetic material when the external magnetic field strength is zero. An excellent superparamagnetic material should have zero or negligible remanence in the absence of a magnetic field. Figure 1 compares the hysteresis loops of a ferromagnetic material with those of magnetite nanoparticles synthesized in our laboratory using the co-precipitation method [19]. The exceptional ability of superparamagnetic particles to exhibit this characteristic property after the removal of the external magnetic field enables them to maintain colloidal stability and avoid agglomeration [20]. Particle shapes and sizes are major determinants for superparamagnetic materials; the particle size should fall in the range of 1–40 nm [20,21]. Spherical nanomaterials are desirable for wastewater treatment because of their comparative high kinetic energy and corresponding ease of clearance [22]. The strength of functionalized magnetic nanoparticles lies in the saturation magnetization (Ms) of the composite; a function of coercivity which is a determinant for recovery and downtime in a reaction medium. To juxtapose metals and metal oxide nanocomposites, metals have relatively higher saturation magnetization but are toxic and extremely sensitive to oxidation; in contrast, metal oxides are less sensitive and give a stable magnetic response [23,24].

Several conventional technologies had been developed for the purification of water. However, the challenge in water purification is the development of a technology that can effectively remove contaminants simultaneously (inorganic, organic and pathogens) and reduce their concentration to ultra-low levels from wastewater [25]. Conventional adsorbents, such as activated carbon, clays, silica beads and biosorbent, have limitations in their permeability, selectivity, temperature dependence, pH dependence and secondary waste [26,27,28]. Other methods of water treatment techniques include reverse osmosis, chemical precipitation, catalytic reduction, ion exchange, electrodialysis and membrane filtration. However, most of these methods are practically expensive, result in the wastage of water, are singly less efficient, are highly specific and have the problem of membrane fouling [29,30,31,32,33]. The development of an all-inclusive water detoxifier is a major challenge in the water industry. Technologies that have been developed are expensive, unscalable and generate disinfection by-products, which is a risk concern for human and environmental health [21].

### 1.2. Smart Magnetic Oxide Nanocomposites for Wastewater Treatment

“Smart” nanocomposites are “stimuli responsive” or “environmentally sensitive” composites [34]. They can undergo physical or chemical changes as responses to small external stimuli in environmental conditions (Figure 2). The band gap value of a material is a measure of the energy required to move an electron from the valence to the conduction band [35]. Therefore, it is desirable for solar-responsive nanocomposites to have low band values for the effective harvesting of solar energy. Figure 2 shows the pictorial representation of a temperature-responsive smart nanocomposite that selectively adsorbs Cu^2+^ and Pb^2+^. The removal of both cations from the aqueous solution would be dependent on the temperature of the medium, assuming the adsorptions of Cu^2+^ and Pb^2+^ on the magnetic oxide nanocomposite are selectively viable at 30 °C and 50 °C, respectively. A solar-responsive composite would enhance the capture and utilization of solar energy for the autoregulation of the system’s temperature, thus allowing for the auto-system adsorption of both cations. The adsorption of contaminants and pollutants from aqueous media using an engineered adsorbent is a facile, robust and industrially scalable method for wastewater treatment. It is a chosen route for the removal of pollutants from wastewater as opposed to photodegradation, which results in the production of secondary pollutants such as carbon (IV) oxide and hydrogen peroxide [36]. Furthermore, both adsorbent and adsorbates are recoverable and re-useable. Adsorption parameters can be used to define the adsorption process. Physisorption is a reversible reaction, while chemisorption involves the formation of new bonds and is not easily reversible. A recyclable adsorbent should be patterned towards a physisorption process for the easy recovery of adsorbates and re-use of the adsorbent. The working principle of a solar-responsive material is dependent on its ability to entrap solar energy and selectively adsorb ions from solution. 

The adsorption process and efficiency are largely dependent on the temperature and pH of the reaction medium. The design of solar-driven superparamagnetic composites would enhance the active capture and harvest of 95% of the untapped energy from the visible region of solar energy for wastewater treatment [37]. This would save on energy and cost for industrial designs. Solar auto-moderated nanocomposites for wastewater treatment would also enhance the assessment of the water–energy nexus in bioscience and technology. Magnetic oxide nanocomposites have the advantage over other nanocomposites as they can be re-used upon the magnetization and desorption of sorbates, thus saving on cost, time and the elimination of secondary pollutants [38].

This review hopes to take a cursory look at the magnetic oxide core-shell for wastewater treatment. Much focus will be given to the magnetic properties of the nanocomposites and their effectiveness as an adsorbent for remediating wastewater. Adsorption parameters as pointers for the development of recyclable sorbents and the solar responsiveness of magnetic nanocomposites will be assessed. Discussion on the potentials of the inclusive modeling of green nanocomposites for wastewater treatment will be expounded, while the cytotoxicity of magnetic oxide nanoadsorbents will be raised.

## 2. Literature Review

The unprecedented growth of nanotechnology and novel properties of magnetic nanoparticles have given rise to wide applications in various industries. Magnetic nanoparticles are adaptable for different biological and environmental media due to their sizes, which are comparable to those of a cell (10–100 µm), a virus (20–450 nm), a protein (5–50 nm) or a gene (2 nm wide and 10–100 nm long) [39]. Furthermore, the relatively small sizes of nanoparticles coupled with functionalization prevent agglomeration and enhance circulation, making the particles less easily recognized by the body’s biological particulate filters [23]. The synthesis of nanoparticles is the first and most important step in the process of water treatment through engineered nanocomposites [6]. Nanoparticle synthesis methods can be categorized into two broad categories: the first is the breakdown (top-down) method, which involves the application of an external force to a solid, breaking it up into smaller particles. The bottom-up method produces nanoparticles starting from atoms of gas or liquids based on atomic or molecular condensations. The top-down approach can be further divided into dry and wet grinding. The wet process is suitable for preventing the condensation of the formed nanoparticles as compared to the dry method. The bottom-up approach, on the other hand, can be further grouped into gaseous phase methods and liquid phase methods. Desirable morphology, usage and properties, among other qualities, are fundamental determinants for the choice of method. Typical methods for the synthesis of magnetic nanoparticles include the sol-gel reaction [40], sonochemical methods [41], hydrothermal synthesis [42], microwave heating, laser and flame spray pyrolysis [43], co-precipitation [44], the green or biological route [45], polyol methods [46,47], the thermal decomposition method [48], solvothermal methods [49], micro-emulsion [50], combustion synthesis the and oxidation method [51]. Amongst all reported methods, the co-precipitation of iron oxides from Fe^2+^ and Fe^3+^ in stoichiometric amounts has been reported and adapted by different authors. This method is facile, well established and is largely scalable [52]. The pH and ionic strength of the precipitating solution are fundamental to the control of the size of the nanoparticle; an increase in both parameters would correspondingly amount to small nanoparticles [53]. Table 1 gives a comparative summary of preparation methods for magnetite nanoparticles.

Considering the possible negative implications of exposure to toxic elements in environmental and biological media, a high level of precision and accuracy is needed for the release of magnetic oxide nanoparticle moieties in vivo. To inhibit bursts and leakages, the superparamagnetic particles are coated to impede undesirable potentials. Figure 3 shows the schematic representations of a coated and functionalized magnetic oxide nanocomposite; the magnetic core bears the coating layer and functionalizers (shells), thus supplying magnetic properties to the composite. The primary coating layer reduces the kinetic energy between the magnetic nanoparticles and brings the stability and improved surface area necessary for the adsorption of pollutants. Coating strategies depend on the initial particle surface and intended usage. The rate-determining steps are bio-functionality, compatibility, stability in a broad pH range and ionic strength [62]. Coating materials include: organic–inorganic shells (e.g., Fe_3_O_4_@SiO_2_-NH_2_) [63]; organic molecules (e.g., EDTAD-Fe_3_O_4_ [64], Fe_3_O_4_@C [65]); polymers (e.g., Fe_3_O_4_@APS@AA-Co-CA) [66]; metal oxides (e.g., MgO-Fe_3_O_4_) [40]; surfactants (Fe_3_O_4_-oleic acid) [44]; inorganic molecules (e.g., Fe_3_O_4_@SiO_2_) [67]; and multi-shells (Fe_3_O_4_@SiO_2_@CS@pyropheophorbide) [68]. Silica coating is primarily via a sol–gel process by the in situ hydrolysis and condensation of silicon alkoxides under a basic environment, forming an inorganosilicate or organosilicate layer on the magnetic oxide nanoparticles, which is rich in silanol groups, a viable covalent coupler for other functional groups [69]. Synthetic methods for some magnetic oxides nanocomposites adsorbents and the influences of coating and functionalizing moieties on their properties for wastewater detoxification are presented in Table 2. 

### 2.1. Preparation Methods for Magnetic Nanoparticles and Composites

Several methods of preparing magnetic iron oxides (MIONs) have been reported by researchers and some selected works are listed in Table 1 and Table 2. Amongst all reported methods of synthesis, the co-precipitation of metal oxides from aqueous solutions with a base under an inert atmosphere is a well-established and scalable route. A comparative study of methods for the preparation of superparamagnetic iron oxide nanoparticles (SPIONs) showed that 90% of SPIONs are prepared by chemical methods; physical method routes make up 8%, with the remaining 2% being biological methods [79]. The study further reported that, amongst chemical methods, co-precipitation occupied a choice position at 28%, followed closely by hydrothermal methods at 26% and micro-emulsion 20%, while the other methods trail by a wide margin. For the biological methods, the protein-mediated route was 60%, while plant-mediated synthesis had a lean share of 2%.

Ge Fei and co-workers produced polymer-modified magnetic oxide composites via the co-precipitation chemical route with sustainable adsorptive strength after four desorption and re-use cycles [70]. The same adsorbent was effective for the adsorption of crystal violet, methylene blue and alkali blue dyes from solution, reaching equilibration after 45 min, and methylene blue had maximum adsorption [66]. In the production of gold-coated magnetic iron oxide nanoparticles, the higher the ratio of Fe(II), the bigger the size of the nanoparticles [80]. The co-precipitation method was used to obtain well-dispersed, water-soluble and biocompatible iron oxide nanoparticles [44,81]. Size control was optimized using a surfactant and at low temperature with a pH range of 9–11; the co-precipitation method can produce fine, high-purity particles of single and multicomponent metal oxide [82,83]. The reaction temperature is also an important consideration during synthesis. Liu and co-workers tuned the particle size of Fe_3_O_4_ nanoparticles by varying the heating temperature and reaction time [48]. Conventionally synthesized magnetic iron oxide nanoparticles (Fe_3_O_4_) produced via the co-precipitation route had a high surface area value of 114 m^2^/g and were chemically stable, with leaching percentages of 0.05% and 0.90% using 0.01M and 1M HCl, respectively, after 72 h [63]. Xu et al. reported Fe_3_O_4_ nanoparticles treated with EDTA dianhydride for effective divalent cation adsorption, maintaining optimum adsorptive strength after three cycles of desorption [64]. Superparamagnetic maghemite (γ-Fe_2_O_3_) nanoparticles produced via the flame spray pyrolysis route resulted in particle sizes distributed within the range of 3.6 nm to 15.7 nm and a specific surface area of 79.4 m^2^/g [71]. Using this adsorbent, the adsorption of copper ions was more favorable at room temperature, while lead ions were effectively adsorbed at 45 °C.

Phosphate ions were effectively adsorbed to the tune of 95% after 5 min when using magnetic oxide nanoparticles produced via the micro-emulsion route; the adsorbent was separated after equilibration from solution within a maximum of 3 min [50]. Maghemite nanoparticles (γ-Fe_2_O_3_) prepared via the sol-gel method had a very low saturation magnetization (Ms) value of 3.3 emug^−1^; the surface area was 178 m^2^/g while the point of zero charge (PZC) was 6.3 and the nanoparticle had a 90% removal rate for Cr(VI) [84]. Aivazoglou et al. reported a facile microwave-assisted synthesis of iron oxide nanoparticles using the biological environment (β-cyclodextrin); the resulting magnetic nanoparticles with a particle size of about 14.75 nm were reproducible within an average of 4 min and had a faceted morphology [85]. The chemical precipitation route was used to synthesize Fe_3_O_4_-PEDOT (poly (3,4-ethylene dioxythiophene)) in a reaction time of 2 h; particle sizes were between the range of 12 to 20 nm. The composite was 95% efficient for heavy metal adsorption, maximum adsorption for the cations was seven times higher than for chitosan coated Fe_3_O_4_ nanoparticles [75]. Sun et al. proposed a facile green hydrothermal method from a single Fe(III) precursor using sucrose as both a capping and stabilizing substrate; reaction time was 2 d; both particle sizes and saturation magnetization were widely distributed between 4.2 and 17.1 nm and 14.82 and 29.55 emug^−1^, respectively [42]. Recently, Parveen et al. produced 1% iron oxide using tannic acid in an alkaline medium; the whole reaction lasted for 30 min [86]. However, the magnetic property of the fabricated iron oxide nanoparticles was not reported. This, however, seems a facile and fast method that can be adopted for other metals using other organic acids. Mandel et al. combined both co-precipitation and sol–gel methods for the continuous synthesis and modification of superparamagnetic nanocomposite microparticles using a Y connection reactor and a magnetic drum for the separation of the produced nanoparticles. The modification with thiol groups reduced the saturation magnetization (Ms) from 30 emug^−1^ to 22 emug^−1^; the composites’ size was 20 µm with a surface area of 75 m^2^/g [87]. There was no reported kinetics study for this work. Zinc ferrite aerogel through the epoxide sol–gel method was used to engineer nitrate-zinc ferrite and chloride-zinc ferrite aerogels; the surface areas of the nitrate aerogels were 371 m^2^/g and 349 m^2^/g, with an average size of 15 nm [88]. Further studies and characterization of the prepared aerogels were not reported. Magnetic nanoparticles exhibit the best characteristic properties in the size range of 10–20 nm; smaller sizes of less than 12 nm would require a large external magnetic field for recovery and separation [38,52]. Microwave heating enhances the synthetic process because, while the traditional method relies on conduction and convection for heat transfer, microwave radiation heats through the more efficient dielectric. The heating process is dependent on the analyte and its matrix to absorb microwaves and convert them to heat; generally, more polar solvents, reagents and catalysts are more efficiently heated [89]. Tsuji et al. have also reported a better control of size and dispersible crystalline nanostructures [90]. The purity of the product is largely dependent on microwave power and reaction time [85,91,92]. Aivazoglou et al. produced magnetic iron oxide nanoparticles in a biocompatible organic environment via microwave-assisted method, starting with polyethylene glycol (PEG) and β-cyclodextrin in the presence of ammonia solution [85]. The resulting magnetic nanoparticles were produced within a record time range of 2.5 to 5 min; the microwave power was between 400 W and 800 W. The particle sizes were between the range of 10.3–19.2 nm, while the saturation magnetization (Ms) was 70 emug^−1^. Kombaiah et al. juxtaposed the conventional heating method (CHM) with the microwave-assisted heating method using okra extract in a green synthesis of CoFe_2_O_4_ nanoparticles. Single-phase crystalline particles were evident within 15 min, in contrast to 1000 °C at 3 h needed for the CHM [93]. De Matteis and co-workers produced an ultrathin MgO coating and magnetite superparamagnetic nanocomposites using the co-precipitation and sol–gel methods [40]. Co-precipitation was used to synthesize the magnetite nanoparticles, while the sol–gel method was used to produce the MgO coating; they inferred that the MgO coating was more effective than SiO_2_.

### 2.2. Magnetic Cores for Magnetic Oxide Nanocomposites

The magnetic strength of the magnetic composite is largely dependent on the magnetic susceptibility of its composition. The Curie temperature of a ferrimagnetic material is the temperature above which it loses its permanent magnetic properties. Iron, cobalt and nickel are the only elements that in metal form have Curie temperatures above room temperature; the Curie temperatures for cobalt, iron and nickel are 1388 K, 1043 K and 627 K, respectively [94]. As such, all magnetic materials should contain one of these elements for sustainable magnetization.

Several researchers have doped magnetic iron oxide nanoparticles to produce novel magnetic cores. Haw and co-workers designed a nanocomposite of CoFe_2_O_4_-TiO_2_ using co-precipitation and hydrothermal routes for the magnetic core and composite, respectively. They effectively used the nanocomposite for the adsorption of methylene blue. However, the composite had a low saturation magnetization (Ms) value of 0.795 emug^−1^ and coercivity of 311.98 G [95]. This is very low compared to the theoretical Ms value of 57.1 ± 1 emug^−1^ for bare Fe_3_O_4_ [96]. The coercivity value of 311.98 G implies that the composite would require more magnetic strength for the separation and recovery of the composite. The researchers attributed the recorded low saturation magnetization (Ms) value for the composite to the presence of diamagnetic TiO_2_. Furthermore, the absence of a coating layer between the magnetic core and TiO_2_ electrovalent linkage would probably have contributed to the low magnetic susceptibility of the composite.

Wu et al. studied the effect of rare-earth substitution on the structural and magnetic properties of cobalt ferrite using Pr^3+^, Sm^3+^, Tb^3+^ and Ho^3+^ via a hydrothermal route [72]. All rare-earth-substituted metals resulted in a decrease in the nanomaterial’s saturation magnetization (Ms). The bare CoFe_2_O_4_ had an Ms value of 82.2 emug^−1^, while Pr(III)-, Sm(III)-, Tb(III)- and Ho(III)-substituted composites had Ms values of 62.5 emug^−1^, 60.2 emug^−1^, 58.4 emug^−1^ and 58.8 emug^−1^, respectively. However, the reported saturation magnetization values should be high enough for magnetization from an environmental medium. Remanent magnetization (Br) decreased for all substituted rare-earth metals; the same was reported for coercivity (Hc). CoFe_2_O_4_ had Hc value of 576 Oe while Br was 20.7 emug^−1^. The rare-earth-substituted composites had values within the range of 240 to 375 Oe for coercivity and 10.2 to 17.3 for remanence. This shows that rare-earth substitution promotes the production of soft magnetic composites that would require less magnetic field strength for magnetization while it reduces the remanence value, which contributes to the agglomeration of composites in environmental media.

Mattila et al. produced cobalt nanoparticles starting with CoCl_2_ powder. The magnetic nanoparticles were coated with carbon and functionalized separately using both 3-aminopropyltriethoxysilane (APTES) and 3-mercatopropyltrimethoxysilane (MPTS) [97]. The stability test of the composite was conducted using Inductively Coupled Plasma-Mass Spectrometry to access cobalt (Co) content in Milli-Q water samples for a maximum of 35 d. The values of leached Co obtained were negligible. Magnetic properties for the cobalt nanoparticles were not reported. Rao et al. produced small magnesium-substituted cobalt ferrite nanoparticles of 4 nm via the sol–gel chemical method [98]. The Curie temperature of the produced nanoparticles was in the range of 446–476 °C; the saturation magnetization (Ms) range was 67.1–84.6 emug^−1^, while coercivity was quite high at 885 Oe. Recently, Santos and co-workers produced a binary magnetic core with nickel-cobalt oxide nanoparticles (NiCo_2_O_4_) using Ni(NO_3_)_2_·6H_2_O and Co(NO_3_)_2_·6H_2_O [99]. The magnetic core was effectively deposited on reduced graphene oxide to form a nanostructure with a mean particle size in the range of 8–10 nm. The surface area was 108 m^2^/g, while saturation magnetization was 21.9 emug^−1^.

### 2.3. Protective or Coating Shells for Magnetic Oxide Nanoparticles

Magnetic oxide nanoparticles (MIONs) are usually coated to protect the core magnetic component from degradation and to enhance stability in aqueous media; the protective covering is normally the sacrificial layer during thermalizing for further functionalization. Several authors have adopted silicon(IV) oxide in the form of tetraethyl orthosilicate (TEOS) for surface coating. Silicon(IV) oxide is commonly adopted because of its chemical inertness, hydrophilicity, non-toxicity and the ease of further modification [69,100]. Polymer-stabilized magnetic iron composites are achieved by heating at a high temperature in an oxygen-free environment. Carbon can effectively protect the Fe_3_O_4_ sphere from corrosion by an acidic medium because its dense structure can block the penetration of hydrogen ions, it also adds very little to the mass of the composite [11,69,101]. The large molecular weight of polymer enhances the surface area of the composite, which is a viable property for biological and environmental applications.

Increasing the silica shell to 56 nm reduced the Ms value to 3 emug^−1^ from an initial value of 61 emug^−1^ [102]. Im et al. reported a very low Ms value of approximately 0.3 emug^−1^ for silicon(IV) oxide-coated magnetic nanoparticles with a diameter of 700 nm [103]. In the work of Zou et al., bare Fe_3_O_4_ particles had a value of 69 emug^−1^; but on increasing the TEOS content to 1.6 mL, the magnetization value reduced to 8.2 emug^−1^ [100]. Furthermore, the composite size exceeded 100 nm, which limits their biological use. Hydrazine was used as a catalyst to prevent the oxidation of Fe_3_O_4_ in place of an inert gas at a temperature of 90 °C. Ge et al. produced magnetic oxide nanoparticles in a core-shell-shell arrangement (Fe_3_O_4_@APS@AA-co-CA); the uncoated magnetic oxide nanoparticles had a saturation magnetization value of 79 emug^−1^, while the composite had a reduced value of 52 emug^−1^ [70]. The Fe_3_O_4_@SiO_2_@NH_2_ composite by Wang et al. [63] had an Ms value of 68.0 emug^−1^ for uncoated Fe_3_O_4_, while the composite Ms value was reduced to about half, to 34 emug^−1^. The particle size increased from 12.09 nm to 13.36 nm for both the uncoated Fe_3_O_4_ and the composite. Replacing hydrophilic iron oxide nanoparticles with organophilic ones greatly improved the magnetic response of silica-loaded superparamagnetic iron oxide nanoparticles. Varying the ratio of TEOS and iron oxide controlled the size of the iron oxide-silica colloid via the Stober process [103]. The TEM analysis (Figure 4) of polyacrylamide-functionalized magnetic oxide carbon-coated nanoparticles showed a homogenous distribution of the particle in the matrix; the composite also showed resistance to leaching in an acidic medium [11]. The polyacrylamide coating on the composite helped formed a dense anti-corrosive layer on the particles, as evident in the SEM image (Figure 4). Though with a very low surface area of 8.3 m^2^/g, the composite showed improved adsorption of metal cations at room temperature. The optimum pH was 6 and the equilibration time was 90 min. Organic surfactants, such as oleic acid or trisodium citrate, have been used to control the size and obtain stable magnetic nanoparticle dispersion. Trisodium citrate played dual functions as a reducing agent and capping stabilizer [52]. Recently, Guivar and co-workers effectively used titanium(IV) oxide to form a protective photocatalytic layer on maghemite nanoparticles [104]. The composite had high sorption for As(III) and As(V) ions after 20 h and was effective within a wide pH range; the adsorbent was recoverable with a permanent magnet within less than 10 min. Comparatively, it can be inferred that organic coatings have a lower reducing effect on composites’ magnetic properties, while inorganic coatings tend to greatly reduce the Ms value. The aggregation and agglomeration of nanoparticles arising from the attractive magnetic force between the particles, which limit their surface interaction with adsorbates, were better curtailed using inorganic moieties. Further study is needed on the development of novel magnetic oxide nanocomposites with hybridized potentials of both organic and inorganic coatings.

### 2.4. Functionalizers for Magnetic Oxide Nanocomposites

Coated magnetic nanoparticles are often activated with different functional groups based on application. The oxidation of the carbonized shell produces oxidized groups on the magnetic core surface, thus enabling the reaction of alkoxysilanes bearing amino- and thiol-reactive functional groups; silanes are often used for the stabilization and functionalization of magnetic nanoparticles since they form a dense layer which can be covalently linked to metal oxides under mild conditions [97]. Badruddoza et al. exploited the ion exchange capability of phosphonium ligands covalently linked to magnetic iron oxide nanoparticles (PPhSi-MIONPs) for the removal of metal ions from water. It was discovered that the adsorption of As(V) and Cr(VI) were not affected by the presence of chloride, nitrate or sulphate in solution [5]; the cations of interest were substantially adsorbed. Magnetite nanoparticles were progressively capped with both 3-aminopropyl triethoxysilane (APTES) and acryloyl chloride (AC) by Mahdavian and Mirrahimi via a co-precipitation route [96]. The magnetic oxide composite gave a 90.3% yield and was used for the adsorption of Cd(II), Pb(II), Ni(II) and Cu(II). Maximum adsorption was observed for Pb(II) while Cd(II) was the least adsorbed. Manganese oxide-functionalized magnetite (Fe_3_O_4_@MnO_2_) was 80% efficient for the adsorption of metal cations and maintained strong adsorptive strength after five cycles; the equilibration time was 10 min [73]. Yew et al. synthesized spherical magnetite (Fe_3_O_4_) with an average particle size of 14.7 nm via a green synthetic route using seaweed (*Kappaphycus alvarezii*) as a reducing and stabilizing agent [105]. X-ray diffraction patterns confirmed the nanoparticle phase as magnetite and not maghemite; further characterizations were not reported but this facile and environment-friendly green route is promising for preparing nanocomposites for environmental and biomedical applications. Humic acid-coated magnetite nanoparticles (Fe_3_O_4_@HA) via a co-precipitation method effectively removed Hg(II) and Pb(II) to the tune of 99% in an equilibration time of 15 min [74]. The composite was stable in tap water, natural water and acidic and basic media. Composite leaching was also negligible, and the removal efficiency was dependent on the organic content of the adsorbate matrix. The produced nanocomposite (Fe_2_O_3_@TiO_2_@GO) by Guivar et al., with a size of 11.3 nm, was 73% and 85% efficient for the adsorption of As(III) and As(V) [104].

Polydispersity as defined by International Union of Pure and Applied Chemistry is the degree of non-uniformity of the size distribution of particles. It is an indication of the aggregation of the magnetic oxide composites. The higher the polydispersity index, the more dispersed the particles; a value close to zero denotes a monodisperse system. Nidhin et al. used alginate, chitosan and starch polysaccharides as templates for the narrow size production of iron oxide particles. As the ratio of polysaccharides increased, the polydispersity for both alginate and chitosan decreased but showed a disordered format for starch [82]. Si et al. described a facile co-precipitation method to prepare monodisperse magnetite nanoparticles from aqueous solution of Fe(II) salt at room temperature at pH 13. Two (2) water-soluble polymers (polyacrylic acid (PAA) and sodium salt of carboxymethyl cellulose (NaCMC)) were adopted as oxidizing or reducing agents; the particle sizes were in the range of 6.6 nm to 16.7 nm depending on the ratio of both precursors. The composite had a magnetization value of 38 emug^−1^ [106]. Gluconic acid, lactobionic acid and polyacrylic acid were used to achieve the biocompatibility of maghemite nanoparticles via a modified co-precipitation route; the study showed that gluconic acid-coated nanoparticles were more dispersed than the other surfactants. The hydrodynamic size was 90 nm, possibly due to hydrogen bond formation between the carboxyl groups on adjacent surfaces, causing cross-linking between particles [107].

The low cost of ZnO and its high potentials against pharmaceutically active compounds (PhACs) and persistent organic pollutants (POPs) makes modified ZnO a viable catalyst for the detoxification of various organic contaminants [108,109]. However, the photocorrosion of ZnO is a major limitation encountered in the wider application of ZnO due to the loss of Zn^2+^ in the water phase and a subsequent decrease in activity [108]. Wibomo et al. incorporated copper and nickel into Fe-doped ZnO nanoparticles using a co-precipitation route and found out that the dopants not only enhanced the ferromagnetic property of the composite but also changed the lattice constant and optical properties [110]. ZnO nanoparticles were doped with lanthanum via a gel combustion route and were found to be a better cytotoxic material for the photocatalytic decontamination of paracetamol than bare ZnO, and the band gap of the ZnO also showed a marked reduction [111]. Razieh had suggested a simple method for the preparation of nano sized ZnO using ZnSO_4_ and NH_4_OH precursors at a temperature of 60 °C for 8 h; XRD analysis revealed a crystallite size of 30 nm [112]. Hamid and co-workers prepared ZnO via a precipitation method using zinc nitrate as a precursor and KOH as a precipitating agent; the reaction time was not reported but both centrifuging and calcination periods were over 3 h and the produced nanoparticles were within the size range of 20–40 nm [113].

Surface modification can be used to tune the ultraviolet and visible light photo-luminescence properties of ZnO nanoparticles and prevent agglomeration [114]. The high dielectric constant of a solvent can leads to smaller particle size; slow addition is desirable for the reaction kinetics [114]. Ba-Abbad et al. prepared Fe^3+^-doped ZnO nanoparticles by a modified sol–gel method using zinc acetate, oxalic acetate and iron acetate precursors and reported a particle size range of 12–18 nm. Increasing the dopant percentage weight to 1.0 wt% resulted in the reduction of the band gap to 2.65 eV from 3.19 eV. The composite also showed an improved photocatalytic property [115]. Recently, Shoueir et al. designed a hybridized green Fenton line nanocatalyst (DNSA@chitosan@MnFe_2_O_4_) for the photodegradation of methylene blue (MB) [116]. They reported that 3,5-Dinitrosalicyclic acid@chitosan (DNSA@CS) was an effective photodecompositor and achieved a 98.9% degradation of methylene blue in 30 min. The redox ability of the composite facilitated the availability of electrons to improve the efficiency of the photodecomposition process.

Titanium(IV) oxide is chemically inert and widely available; titanium is the 9th most available metal on earth [117]. It responds to UV irradiation and has been adopted as a photocatalyst for the degradation of pharmaceutically active compounds and recyclable photocatalysis applications [95,108]. A major drawback in the application of TiO_2_ as a photocatalyst is the broad band gap within the range of 3.0–3.5 eV, which limits its absorption to the UV region, which is only about 5% of solar light, and the easy recombination of the photoexcited electron–hole pair [37,118]. Transition and noble metal doping, doping with carbon, dye sensitizers and metal impurities are some of the suggested approaches to enhance the photocatalytic properties of TiO_2_ [21,118,119]. Photocatalytic material should have a narrow band gap in the range of 1.23 to 3.0 eV for an optimum harvest of solar radiation [120]. The doping of TiO_2_ with magnetic oxide nanoparticles would help in the narrowing of its band gap value; the enhancement of its surface properties; and inducing magnetic properties which are desirable for the engineering of environment-responsive composites.

### 2.5. Magnetic Oxide Nanocomposites for Wastewater Treatment

The most studied nanosized metal oxides for metal remediation from aqueous media are iron oxides, manganese oxides, aluminum oxides and titanium oxides [121]. Among these, only iron oxides and manganese oxides are ferrimagnetic oxides. Other magnetic oxides, such as nickel and cobalt, are seldom studied for biological and environmental remediation because of their toxic potentials in aqueous media. Iron has the unique ability to exist in various oxidation states: Fe(0), Fe(II) and Fe(III), while the oxidation states from IV to VI are referred to as ferrates [122]. A major problem in the application of magnetic iron oxide nanoparticles (MIONs) is the air oxidation of iron(II). The air oxidation of magnetic iron oxide nanoparticles leads to the loss of magnetism and dispersibility [44]. Both magnetite and maghemite are ferrimagnetic as multi-domain materials (bulk materials); however, magnetite has a larger saturation magnetization, while maghemite is usually more stable in aqueous media [20]. Therefore, an analytical blend and ratio of both Fe(II) and Fe(III) are determinants for stability, saturation magnetization and particle size, shape and morphology. Nano zero-valent iron has a high reactivity towards a broad range of contaminants, including halogenated compounds, nitrates, phosphates, polycyclic aromatic hydrocarbons and heavy metals [123].

Adsorption is a thermodynamic equilibrium surface reaction that involves a net accumulation of a substance at the common boundary of two contiguous phases [124]. A state of equilibrium is achieved in the adsorption process when the concentrations of pollutants on the solid and in the solution become constant. At this state of equilibrium, the relationship between the amount of solid adsorbed and in solution is called an adsorption isotherm. Adsorption isotherms are important for the description of the interaction of adsorbate with adsorbent and are critical in optimizing the use of adsorbents. The presence of competing ions in solution can interfere with the active adsorption of a primary analyte. In the presence of co-existing ions, the adsorption of Cu(II) ions onto Fe_3_O_4_@SiO_2_@NH_2_ slightly decreased; the presence of humic acid had a negligible effect on the overall adsorption process [63]. The presence of SO_4_^2−^, HCO_3_^−^, SiO_3_^2−^ and PO_4_^3−^ in solution did not affect the adsorption of As(III) to magnetic cellulose adsorbent [76].

Othman et al. produced magnetic graphene oxide which had 99.6% efficiency for the remediation of methylene blue from an aqueous medium [125]. The regeneration study was done using 1M acetic acid. Aigbe et al. proposed a novel method for the removal of Cr(VI) using a polypyrrole-functionalized magnetic composite; the magnetic composite had low saturation magnetization and a small surface area of 28.77 m^2^/g, though it was 99.2% effective in the remediation of Cr(VI) from aqueous solution [126]. Hu et al. effectively adopted adsorption and magnetic separation using maghemite for the remediation of Cr(VI) from aqueous media; after equilibration, the spent adsorbent was regenerated using 0.01M sodium hydroxide [84]. The capacity of the adsorbent remained unchanged after six cycles. The regeneration of Fe_3_O_4_@SiO_2_@NH_2_ adsorbent was feasible using a 1 mol/L HCl solution and had satisfactory adsorption after four cycles [63]. Cationic dyes were effectively desorbed from Fe_3_O_4_@APS@AA-co-CA using a mixture of methanol and acetic acid solution (acetic acid 5% *v*/*v*) [66]. The use of 0.01M NaOH as an eluent was effective for the desorption of Cr(VI) from maghemite nanoparticles [84]. The use of 0.1M HCl gave maximum recovery of metal ions after the adsorption of As(III) to a magnetic cellulose adsorbent [76]. Recently, Wanjeri et al. produced a magnetic composite of Fe_3_O_4_@SiO_2_@GO-PEA that effectively adsorbed within a wide pH range [77]. The magnetic composite was used for the adsorption of organophosphorus (OPP) pesticides. The modification of graphene oxide (GO) with 2-phenylethylamine (PEA) made the surface of the magnetic nanocomposite pH independent; the composite was reusable for a 10-cycle period. Mian and Liu synthesized a TiO_2_/Fe/FeC-biochar composite from sewage sludge as a heterogeneous catalyst for the degradation of methylene blue [127]. Chitosan inclusion improved the surface area and mesoporosity of the composite, enabling high catalytic activity in the dye degradation process through H_2_O_2_ activation. Cefotaxime, a ubiquitous antibiotic, was efficiently degraded by 82.48% after 100 min by a bimetallic nanocomposite (Co/Fe/MB) functionalized with alkali-modified biochar [128]. The researchers exploited the free electronic orbit of cobalt for the adsorption of atomic hydrogen and improved the degradation of the dye. The physiochemical and catalytic properties of N-doped metal/biochar were investigated by Mian and co-authors [129]. Composites prepared at a pyrolysis temperature of 800 °C (N-TiO_2_-Fe_3_O_4_-biochar) performed best. The broad band at 3400 cm^−1^ indicated the co-existence of NH and OH functional groups. The band gap of the composite at 1.94 eV would promote the effective harvest of solar energy in the visible region.

There are limited reports on the application of metal oxide composites as remediators for field wastewater. Wanjeri et al. [77] demonstrated the applicability of their composite for raw wastewater treatment by simulating the adsorption of organophosphorus pesticide using water from the Vaal dam and river. They reported 86.9% and 90.1% recovery of the pesticide from the dam and river simulated media, respectively. The removal of Cr(VI) by a polypyrrole magnetic composite was reported to be strongly dependent on the applied magnetic field strength [126]. Recently, Sun et al. [130] engineered a multifunctional iron-biochar composite for the simultaneous removal of toxic elements, inherent cations and hetero-chloride from hydraulic fracturing wastewater in an 8-h equilibration period. The study revealed that the ratio of iron to biochar in the composite had effects on the sorption of the pollutants. Maximum removal for Na, Ca, K, Mg, Sr and Ba cations was less than 30%; the researchers attributed the low sorption to the increased positive charge of the composite by iron loading and the corresponding electrostatic repulsion force on the cations. In addition, 1, 1, 2-trichloroethane had a maximum sorption at 91%, while Cr(VI) and As(V) were removed to the tune of 58.4% and 65.9%, respectively. The recyclability study of the magnetic composite, however, was not reported. The pseudo-second-order rate equation described the reaction kinetics of the pollutants from the wastewater better. The report of Sun and co-workers is a milestone in the investigation of multifunctional magnetic composite viability for wastewater treatment. More studies are needed on the removal of anions, polychlorinated biphenyls (PCBs), endocrine-disrupting chemicals, dioxins, radioactive compounds and other hydrophilic and hydrophobic pollutants for the informed modeling of raw wastewater treatment.

### 2.6. Mechanism of Pollutant Removal by Metal Oxide Nanocomposites

Since most pollutants are not magnetic, the application of a magnetic field to remove pollutants from wastewater treatment plants can help to decrease the time needed for sedimentation [131]. Furthermore, the adsorption of both hydrophobic and hydrophilic pollutants can be enhanced with the application of a magnetic field to direct adsorbents to pollutant hotspots. The adsorption process is mainly moderated by two environmental conditions, which are the temperature and pH of the reaction medium. The mechanism of the clearance of pollutants from an aqueous medium by magnetic oxide nanocomposites is also dependent on the adsorbate, adsorbent and the solution matrix. The nature of the composite in terms of its surface properties, as well as the nature and charge of the analyte of interest, are important considerations. For example, Cr(VI) exists as CrO_4_^2−^ at a pH greater than 6 and as Cr_2_O_7_^2−^ and HCrO_4_^−^ at a pH lower than 6 [132]. The reduction of Cr(VI) to the less toxic Cr(III) has been identified as the route of clearance of Cr(VI) [133]. It has also been reported that the removal of Cr(VI) by magnetic biochar was via reduction by Feᴼ, followed by adsorption via ligand exchange with C-O and N-H functional groups on the iron-biochar nanocomposite [130]. In a pH range of 5.5–12.5, lead ions exist as Pb(OH)_2_ and as Pb(OH)_4_^2−^ at a pH greater than 12.5 [134,135]. Metal(II) ions were removed via surface adsorption and complexation, while inner sphere ligand exchange was responsible for the removal of As(V) [130]. Conclusively, when the pH of the solution is lower than the pH(pzc), the reaction medium becomes acidified and proton formation is favored over the hydroxide group [136]. This makes the nanocomposite’s surface positively charged and promotes the adsorption of anions. This then implies that, for an electropositive nanocomposite with a pH(pzc) greater than the pH of the medium, the adsorption of Pb(II) would be favorable at an alkaline pH, while Cr(VI) adsorption would be more favorable in an acidic pH.

Functionalized composites readily form dentate via their donor groups with metal ions on the surface of the composite; also, anions are adsorbed on the surface of the composite by electrostatic forces. Figure 5 gives an illustrative mechanism for the adsorption of some pollutants on a biochar-capped magnetite nanocomposite which has been functionalized using poly(3,4-ethylenedioxythiophene (PEDOT). The high sorption capacity of chlorpyrifos and parathion organophosphorus pesticides was related to the π-π interaction of the phenyl ring on the adsorbent, with the adsorbate in addition to hydrogen bonding [77]. Toluene, ethylbenzene and xylene were effectively adsorbed on the surface of functionalized multi-walled carbon nanotubes via hydrogen bonds and π-π interactions [137]. The sorption of six named dyes belonging to the indigoid, arylmethane and azo classes of dyes on Fe_3_O_4_@haemogloin were reported to be dominated by electrostatic interactions [12]. Adsorbed analytes are subsequently recovered by eluting with a suitable solvent (Figure 6). The ease of recovery of adsorbed pollutant(s) with a suitable solvent can be used to understand the sorption process. A physisorption process would be easily recovered, while a chemisorption process might be difficult because of potential bond formation between the adsorbate and adsorbent. In Figure 6, the desorption efficiencies of Pb(II) from the magnetic oxide after three cycles using 0.1 mol/L HCl as an eluent were 85.58%, 82.82% and 81.47%, respectively. The consistency of the recovery percentages communicates a physisorption process and a commercially adaptable composite; the complexation of Pb(II) with EDTA was easily dissociated using a dilute concentration of HCl.

The adsorptive properties of some selected magnetic oxide nanocomposites used for wastewater treatment are presented in Table 3 while saturation magnetization and adsorption kinetic parameters presented in Table 4. Adsorption data are useful parameters for the explanation of the adsorption process (Table 3 and Table 4). The thermodynamic quantities of Gibb’s free energy (ΔG), entropy (ΔS) and enthalpy (ΔH) give a quantitative measurement of the randomness of the system and heat involved in a reaction, respectively. For physisorption, enthalpy should fall within the range of 2.1–20.9 KJ mol^−1^, while chemisorption processes are 80–200 KJ mol^−1^) [138,139]. The Langmuir isotherm has been largely adopted by scientists as a modeling tool to describe the adsorption process; the separator factor (R_L_), a Langmuir-dependent parameter, indicates the affinity of the adsorbate to bind to the adsorbent, and a physisorption process must have a value less than unity [140,141]. A largely physisorption process would have an infinitesimal value for R_L_ and vice versa. In the work of Guivar et al. [104], the calculated q_e_ (amount of adsorbate adsorbed on adsorbent at equilibrium) for the pseudo-first order (PFO) was not consistent with the obtained value for adsorption capacity, prompting further extrapolation using the pseudo-second order (PSO), which had good linearity and agreement with the experimental equilibrium capacity. Most authors have reported the pseudo-second order (PSO) as a choice kinetics equation for the interpretation of the adsorption process. Recently, Shikuku et al. [142] reported errors in parameter estimation using a linearized adsorption isotherm. It is therefore imperative to compare both parameters for linear and non-linear forms of adsorption isotherms and their associated errors before statistically determining the goodness-of-fit model, considering the ubiquitous interaction of adsorbent-adsorbate in aqueous media. Several works considered during this review had not reported thermodynamic studies, hence the inability to extrapolate enthalpy, entropy and Gibb’s free energy of the adsorption process. Kinetics, isotherm and thermodynamic data of the reaction process are indispensable tools for industrial scale-up and design.

## 3. Conclusions

Magnetic oxide nanocomposites are eco-friendly adsorbents for wastewater treatment. The coating and functionalization processes reduce agglomeration and promote colloid stability. Furthermore, these nano adsorbents are applicable to range of pollutants, are easily recoverable and re-usable and are largely scalable and relatively cost effective. The synthetic route in addition to the nature of the capping and functionalizing moieties impacts the morphology and magnetic properties of the nanocomposites. Organic capping agents have a lower reducing effect than inorganics on the hysteresis loop of the magnetic oxide nanocomposite. Adsorption is a cheap, facile and robust method for the removal of pollutants from wastewater. However, the efficiency of the process is dependent on both the pH and temperature of the reaction matrix. More research is needed for the development of environmentally responsive and friendly nanocomposites with inherent capabilities to work within a wide pH and temperature range. An environment auto-moderated magnetic nanocomposite would harness natural sources of energy and significantly reduce chemical and engineering costs, while driving a green economy in tandem with the United Nations Sustainable Development Goals.

For better adaptation, more studies still need to be done on developing facile and environment-friendly methods to produce superparamagnetic nanocomposites with large saturation magnetization (Ms); high specific surface area; and anti-corrosive coating on the surface of the magnetic nanostructures, while providing abundant adsorption sites and efficient process design for the simultaneous adsorption and subsequent desorption of ubiquitous pollutants from wastewater. Studies on the optimal saturation magnetization values of magnetic oxide nanocomposites, recovery time and the adsorbate–adsorbent relationship in the presence of a magnetic field during adsorption are desirable for wastewater system engineering. Furthermore, kinetics and thermodynamic properties from pilot studies in the laboratory should be comprehensive and detailed to serve as a modeling tool for industrial scale-up and design. The applied effect(s) of a magnetic field in both biological and environmental media are also desirable for informed policy and environmental frameworks.

## Figures and Tables

**Figure 1 molecules-25-04110-f001:**
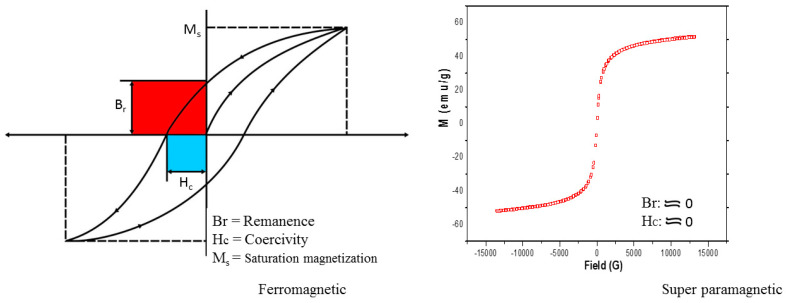
Hysteresis loops for ferromagnetic material and magnetite nanoparticles.

**Figure 2 molecules-25-04110-f002:**
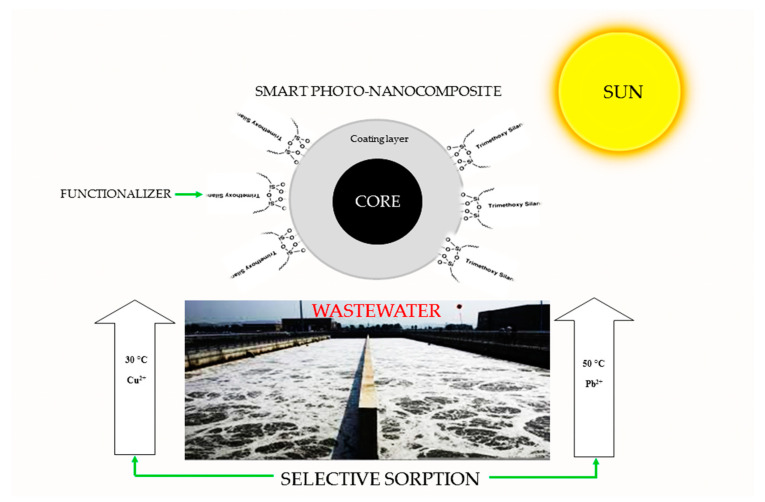
Pictorial representation of a temperature-responsive smart nanocomposite, showing selective adsorptions of Cu^2+^ and Pb^2+^ at 30 °C and 50 °C, respectively.

**Figure 3 molecules-25-04110-f003:**
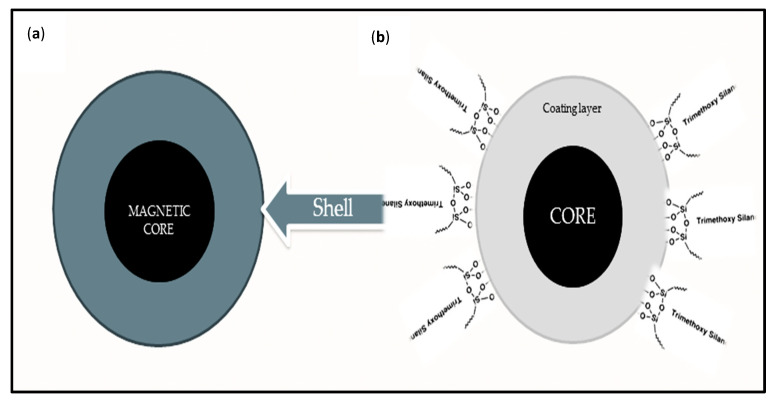
(**a**) Coated and (**b**) functionalized magnetic oxide nanocomposites.

**Figure 4 molecules-25-04110-f004:**
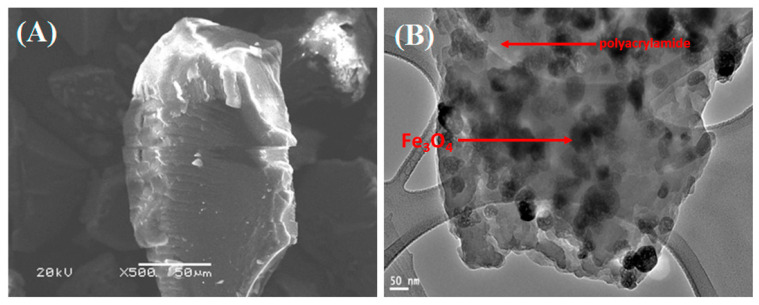
SEM (**A**) and TEM (**B**) images of polyacrylamide-functionalized magnetite, showing a dense anti-corrosive surface and uniform distribution of magnetite nanoparticles in the polyacrylamide capping moiety [11].

**Figure 5 molecules-25-04110-f005:**
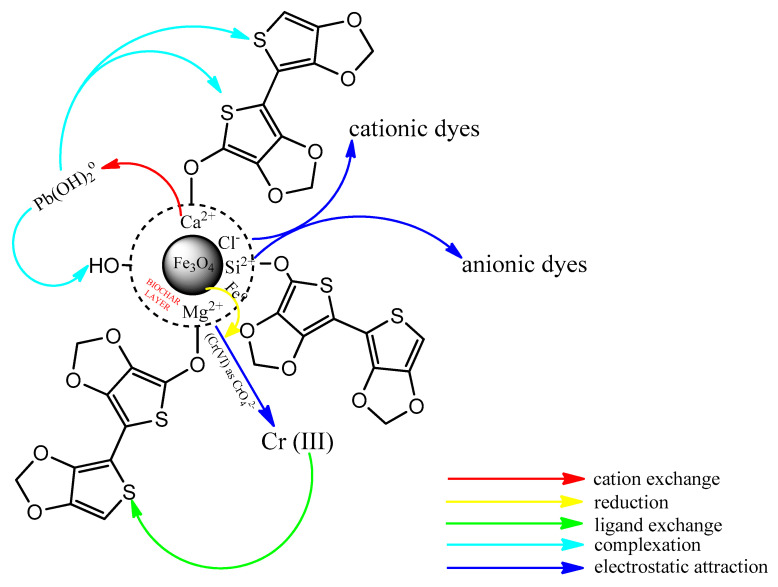
Proposed mechanism for the removal of some pollutants on a biochar-capped and poly(3,4-ethylenedioxythiophene)-functionalized composite [130].

**Figure 6 molecules-25-04110-f006:**
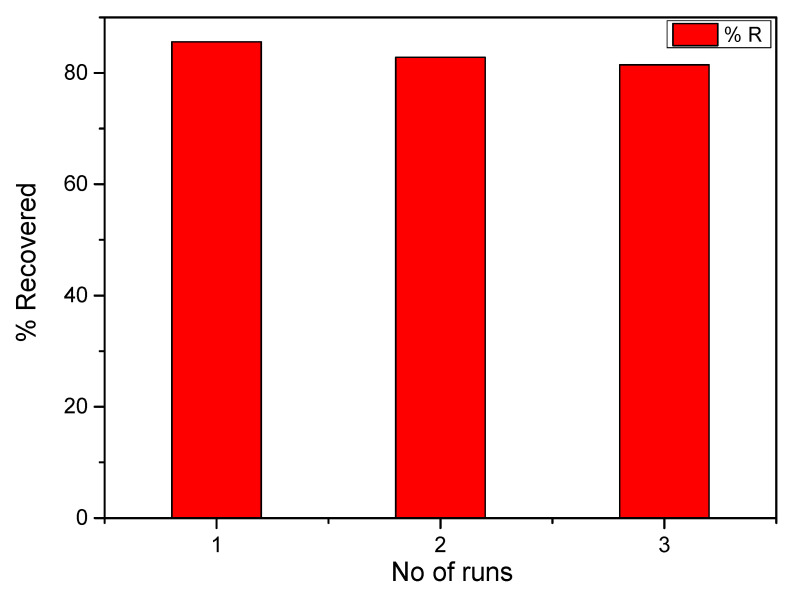
Recovery plot of Pb(II) ions from Fe_3_O_4_@yeast/EDTA composite after three cycles [64].

**Table 1 molecules-25-04110-t001:** Comparative analysis of preparation methods for magnetite nanoparticles.

Nos	Method	Advantages	Disadvantages	* Cost	Ref
**1.**	Hydrothermal	Single step synthesis; high Ms value; highly crystalline particles	Cubic particles; large particle sizes; long reaction time; polydispered size distribution; high reaction temperature	$44.01	[54]
**2.**	Thermal Decomposition	Medium particle sizes; high Ms; short reaction time	Cubic–spherical shapes; polydispered size distribution	$73.84	[55]
**3.**	Co-precipitation	Spherical and small particles; high Ms; low reaction temperature; one-step synthesis; monodispered size distribution	Agglomerated particles	$39.50	[56]
**4.**	Sol-gel	Spherical nanoparticles; small particle sizes	Polydispered size distribution; long reaction time; multi-step reaction process	$99.12	[57]
**5.**	Sonochemical	High Ms; short reaction time; one-step reaction;	Nanocubes; large particle sizes; polydispered size distribution	$84.39	[58]
**6.**	Polyol	Spherical and small particles; moderate Ms; monodispered size distribution	Multi-precursors and reaction steps.	$141.07	[59]
**7.**	Electrochemical	High Ms; small particle size; monodispered size distribution; room temperature synthesis	Prone to impurity; quasi-spherical particles; hydrophobic nanoparticles	$32.55	[60]
**8.**	Microemulsion	Small particle size; monophased product; monodispered size distribution	Agglomerated and cubic particles; multi-precursors	$242.72	[61]

* Particle size: small (2–15 nm); medium (16–25 nm); large (26 nm and above); saturation magnetization (Ms): low (0–30 emug^−1^); moderate (31–69 emug^−1^); high (70 emug^−1^ and above); total cost was calculated based on the cost of (1g or 1L) of primary precursor(s) as obtained from (MERCK South Africa) and energy cost per Kwh of electricity (https://www.globalpetrolprices.com/data_electricity_download.php).

**Table 2 molecules-25-04110-t002:** Synthetic methods for magnetic oxide adsorbents: merits and demerits.

Material	Method of Synthesis	* Functionalizer	Comment	Ref.
Fe_3_O_4_	Co-precipitation	PPhSi	Saturation magnetization (Ms) value of 68 emug^−1^ is close to Ms value of magnetite. Less bulky phosphonium group must have enhanced Ms value. TEM images showed particles were an average of 12 nm and spherical. Morphology is desirable for wastewater treatment.	[5]
Fe_3_O_4_@C	Hydrothermal	polyacrylamide	Cubic particle size in the range of 50–70 nm are limitations for the application of the composite for wastewater treatment. Furthermore, polyacrylamide decreased Ms value from 67 emug^−1^ to 31 emug^−1^.	[11]
Fe_3_O_4_@SiO_2_	Co-precipitation	APTMS	Low Ms value of 34 emug^−1^ is attributed to the bulky silanol group; however, an average particle size of 18 nm is desirable for a superparamagnetic composite.	[63]
Fe_3_O_4_@yeast	Co-precipitation	EDTAD	Particle size, saturation magnetization, and shape were not reported. The presence of EDTAD as a functionalizer is desirable in the complexation and adsorption of Pb^2+^, Ca^2+^, Cd^2+^.	[64]
Fe_3_O_4_@APS	Co-precipitation	AA-co-CA	The cubic shape of the particles can limit its application in wastewater treatment; however, Ms value of 52 emug^−1^ and size of 18 nm are good merits.	[66]
Fe_3_O_4_@APS	Co-precipitation	APS	Average particle size of 15 nm and Ms value of 67 emug^−1^ are desirable for wastewater treatment. Cubic-structured particles can pose limitations during clearance.	[70]
γ-Fe_2_O_3_	Flame spray pyrolysis	-	The comparatively low saturation magnetization of 44.5 emug^−1^ and platelet shape of the particles are quite a limitation. However, average particle size of 10 nm is desirable.	[71]
CoFe_2_O_4_	Hydrothermal	Pr^3+^, Sm^3+^, Tb^3+^, Ho^3+^	High reaction temperature gives little control over the properties of the particles, resulting in quasi-spherical agglomerated particles. Nevertheless, particle size of 9.2 nm and Ms value of 60 emug^−1^ are desirable.	[72]
Fe_3_O_4_	Hydrothermal	MnO_2_	Reported average size of 60 nm and slightly agglomerated quasi-shaped particles are limitations of nanocomposite application in wastewater treatment. Ms value of 40 emug^−1^ should be enough for magnetic recovery.	[73]
Fe_3_O_4_	Co-precipitation	Humic acid	High Ms value of 79.6 emug^−1^ and particle size of 10 nm showed humic acid sparingly affected the magnetic properties of the nanoparticles. TEM images of composites were spherical and showed no agglomeration.	[74]
Fe_3_O_4_	Chemical precipitation	PEDOT	Monodispersed and spherical particles of 11 nm are desirable for wastewater treatment. However, saturation magnetization value of 24 emug^−1^ is relatively small when compared to bare magnetite nanoparticles.	[75]
Fe-oxide	Biological	Cellulose	Reported mixture of rod- and cone-shaped particles with grain size range of 40–110 nm can affect sustained magnetic response and clearance from wastewater.	[76]
Fe_3_O_4_@SiO_2_	Co-precipitation	GO-PEA	Spherical particles with an average size of 22 nm are desirable for wastewater treatment. A major limitation is the sharp decrease in Ms value from 77 to 33 emug^−1^ upon coating and functionalization.	[77]
Fe_3_O_4_@Cu	Hydrothermal	MgO-Cu	Particles were agglomerated though spherical; however, average particle size of 50 nm can be a limitation. Diamagnetic MgO coating reduced Ms value to 29 emug^−1^.	[78]

* RE = (Pr^3+^, Sm^3+^, Tb^3+^, Ho^3+^); APS = 3-aminopropyltriethoxysilane; AA = acrylic acid; CA = crotonic acid; PPhSi = phosphonium silane; GO = graphene oxide; EDTAD = ethylene diaminetetraacetic dianhydride; PEDOT = poly(3,4-ethylenedioxythiophene); PEA = 2-phenylethylamine; APTMS = 3-aminopropyl)-trimethoxysilane; Ms = saturation magnetization.

**Table 3 molecules-25-04110-t003:** Adsorptive properties of selected magnetic nanocomposites for wastewater treatment.

Adsorbent	* Mass (g/L)	Surface Area (m^2^/g)	Adsorbates	Adsorption Capacity (mg/g)	Ref.
**Fe_3_O_4_**	12	105.7	As^5+^	50.5	[5]
			Cr^6+^	35.2	
**Fe_3_O_4_@C/PAA**	0.3	8.2	Cu^2+^	194	[11]
			Ni^2+^	144.3	
			Co^2+^	128	
			Cd^2+^	161	
**Fe_3_O_4_@haemoglobin**	2	12.43	Eriochrome black T	178.6	[12]
			Indigo carmine	104.2	
			Naphthol blue black	114.9	
			Tartrazine	80	
			Erythrosine	178.6	
			Bromophenol blue	101	
**Magnetic Iron oxide**	0.4	142	PO_4_^2−^	40.0	[50]
**Fe_3_O_4_@SiO_2_/NH_2_**	0.4	216.2	Cu^2+^	43.8	[63]
			Pb^2+^	111.9	
			Cd^2+^	37	
**Fe_3_O_4_@yeast/EDTAD**	1	-	Pb^2+^	88.16	[64]
			Ca^2+^	27.19	
			Cd^2+^	40.70	
**Fe_3_O_4_@APS/AA-co-AA**	1	-	Methylene blue	124	[66]
			Crystal violet	180.5	
			Alkali blue	17.8	
**Fe_3_O_4_@APS/AA-co-CA**	1	-	Cd^2+^	29.6	[70]
			Zn^2+^	43.4	
			Pb^2+^	166.1	
			Cu^2+^	126.9	
**γ-Fe_2_O_3_**	0.2	79.4	Pb^2+^	68.9	[71]
	0.1		Cu^2+^	34.0	
**CoFe_1.9_RE_0.1_O_4_**	0.5	82	Congo red	152.0	[72]
**Fe_3_O_4_@MnO_2_**	1	118	Cd^2+^	53.2	[73]
**Fe_3_O_4_@humic**	0.1	64	Cu^2+^	46.3	[74]
			Cd^2+^	50.4	
			Pb^2+^	92.4	
			Hg^2+^	97.7	
**Fe_3_O_4_@PEDOT**	-	-	Ag^+^	3016	[75]
			Hg^2+^	3200	
			Pb^2+^	3105.9	
**Fe-oxide@cellulose**	1	-	As^3+^	92.95	[76]
**Fe_3_O_4_@SiO_2_/GO-PEA**	1	133	Chlorpyrifos	87	[77]
			Malathion	74	
			Parathion	86	
**Fe_2_O_3_@TiO_2_/GO**	0.6	82	As^3+^	110.4	[104]
			As^5+^	90.2	
**Fe_3_O_4_-polypyrrole**	4	28.77	Cr^6+^	208.7	[126]
**Fe_3_O_4_@MWCNT-KOH**	0.4	662.1	Toluene	63.34	[137]
			Ethylbenzene	249.44	
			Xylene (meta)	227.05	
			Xylene (otho)	138.04	
			Xylene (para)	105.59	
**Adsorbent**	*** Mass (g/L)**	**Surface area (m^2^/g)**	**Adsorbates**	**Removal efficiency (%)**	**Ref.**
**Fe_3_O_4_@Cu/MgO-Cu**	**0.32**	**90.1**	***Escherichia coli*** ***Staphylococcus aureus***	**99%**	[78]
**CoFe_2_O_4_@TiO_2_**	4	44.38	Methylene blue	98.89%	[95]
**DNSA@CS/MnFe_2_O_4_**	0.06	219	Methylene blue	98.9%	[116]
**TiO_2_/Fe/FeC-biochar**	-	267.30	Methylene blue	94.4%	[127]
**Co/Fe/MB**	0.8	262	Cefotaxime	82.48%	[128]
**N-TiO_2_-Fe_3_O_4_-C**	1	-	Methylene blue	99%	[129]
**Fe-biochar**	2	138.4	Na^+^,Ca^2+^,K^+^,Mg^2+^,		[130]
			Sr^2+^,Ba^2+^	<30%	
			Cr (VI)	58.4%	
			As (V)	65.9%	
			1,1,2-trichlorethane	91%	

* Mass = adsorbent mass; RE = (Pr^3+^, Sm^3+^, Tb^3+^,Ho^3+^); APS = 3-aminopropyltriethoxysilane; AA = acrylic acid; Co/Fe/MB = cobalt/iron/modified biochar; PAA = polyacrylamide; GO = graphene oxide; DNSA@CS = 3,5-dinitrosalicyclic acid/chitosan; EDTAD = ethylene diaminetetraacetic dianhydride; PPhSi = phosphonium silane; PEA = 2-phenylethylamine; PEDOT = poly (3,4-ethylenedioxythiophene; N-TiO_2_-Fe_3_O_4_-C = nitrogen-titanium(IV) oxide-magnetite-biochar; CA = crotonic acid; TiO_2_/Fe/FeC-biochar = titanium(IV) oxide/iron/iron-chitosan-biochar; MWCNT = multi-walled carbon nanotube.

**Table 4 molecules-25-04110-t004:** Kinetics parameters of selected magnetic oxide nanocomposites for wastewater treatment.

Magnetic composite	^a^ pH	^c^ Time	Kinetics	Isotherm	^b^ Ms (emug^−1^)	Ref.
**Fe_3_O_4_-PPhSi**	6.0	0.83 h	PSO	Langmuir	68.2	[5]
**Fe_3_O_4_@C@PAA**	-	1.5 h	PSO	Langmuir and Freundlich	31	[11]
**Magnetic iron oxide**	-	0.5 h	-	Langmuir	-	[50]
**Fe_3_O_4_@SiO_2_/NH_2_**	6.0	24 h	-	Langmuir	34	[63]
**Fe_3_O_4_@yeast/EDTAD**	-	2 h	PSO	Langmuir and Freundlich	-	[64]
**Fe_3_O_4_@APS/AA-co-CA**	3.5	0.75 h	PSO	Langmuir	52	[66]
**Fe_3_O_4_@APS@AA-co-CA**	3.5	0.75 h	PSO	Langmuir and Freundlich	52	[70]
**γ-Fe_2_O_3_**	6.3	3 h	PSO	Langmuir and Freundlich	44.5	[71]
**CoFe_1.9_RE_0.1_O_4_**	-	1.5 h	PSO	Langmuir model	60	[72]
**Fe_3_O_4_@MnO_2_**	3.7	0.5 h	PSO	Langmuir	40	[73]
**Fe_3_O_4_@humic**	6.0	0.25 h	-	-	79.6	[74]
**Fe_3_O_4_@PEDOT**	-	24 h	-	-	24	[75]
**Fe-oxide@cellulose**	7.8	3 h	PSO	Langmuir	57.2	[76]
**Fe_3_O_4_@SiO_2_/GO-PEA**		0.25 h	PSO	Sips	33	[77]
**CoFe_2_O_4_@TiO_2_**	-	6 h	PFO	-	0.211	[95]
**Fe_2_O_3_@TiO_2_/GO**	8.6	20 h	PSO	Langmuir	60	[104]
**DNSA@CS@MnFe_2_O_4_**	-	0.5 h	-	Langmuir-Hinshelwood	29.95	[116]
**Fe_3_O_4_-polypyrrole**	-	24 h	-	Langmuir	14	[126]
**TiO_2_/Fe/FeC-biochar**		5 h	-	-	47.60	[127]
**Co/Fe/MB**	-	1.6 h	PSO	-	-	[128]
**N-TiO_2_-Fe_3_O_4_-C**		5 h	-	-	19.26	[129]
**Fe-biochar**		**8 h**	**PSO**	**-**	**-**	[130]

^a^ pH = point of zero charge; ^b^ Ms = saturation magnetization; PSO = pseudo second order; PFO = pseudo first order; ^c^ Time = equilibration time.

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
