# Peer review of "Multifunctional Magnetic Oxide Nanoparticle (MNP) Core-Shell: Review of Synthesis, Structural Studies and Application for Wastewater Treatment"

_molecules, 2020, doi:10.3390/molecules25184110_

Round 1

Reviewer 1 Report

This is an interesting review which deals with oxide nanoparticles, their synthesis and applications. I found this work very interesting, however, there are three main concerns that should be address before accept this manuscript for publication:

  1. Please check minor grammatical mistakes that were underlined in yellow in the attached version of this manuscript.
  2. I suggest also to check the format of the references, some examples were also underlined in yellow in the respective section, but it is important to check all of them.
  3. Many of the references are quite old, please try to update some of the references.

After these observations are addressed, I think this manuscript can be published in molecules.

Author Response

Reviewer #1

This is an interesting review which deals with oxide nanoparticles, their synthesis and applications. I found this work very interesting, however, there are three main concerns that should be address before accept this manuscript for publication:

Please check minor grammatical mistakes that were underlined in yellow in the attached version of this manuscript.

Response

Grammatical mistakes have been checked carefully in the manuscript and as corrected in the attached version of the manuscript.

I suggest also to check the format of the references, some examples were also underlined in yellow in the respective section, but it is important to check all of them.

Response

The references have been checked for proper formats, punctuation and abbreviations.

Many of the references are quite old, please try to update some of the references.

Response

The manuscript has been updated with current articles as advised.

Reviewer #1

This is an interesting review which deals with oxide nanoparticles, their synthesis and applications. I found this work very interesting, however, there are three main concerns that should be address before accept this manuscript for publication:

Please check minor grammatical mistakes that were underlined in yellow in the attached version of this manuscript.

Response

Grammatical mistakes have been checked carefully in the manuscript and as corrected in the attached version of the manuscript.

I suggest also to check the format of the references, some examples were also underlined in yellow in the respective section, but it is important to check all of them.

Response

The references have been checked for proper formats, punctuation and abbreviations.

Many of the references are quite old, please try to update some of the references.

Response

The manuscript has been updated with current articles as advised.

Reviewer 2 Report

Dear authors,

The manuscript Molecules-881517 is a well-organised review paper about multifunctional magnetic oxides nanoparticles (MNP’s) core-shell, that reports interesting aspects of MNP’s synthesis, structural studies and application for waste-water treatment. This paper can be slightly improved (please, see my comments below).

This paper is suitable for publication in Molecules Journal after minor revision.

Comments:

Lines 148-154: It would be good to add a table with brief description of each typical methods for the synthesis of magnetic nanoparticles and outlining advantages and disadvantages of these methods, maybe also include cost assessment.

Lines 347-350: Did you get permission from authors [10] to use their figure 1 in your paper? I think it is very important to get permission from the authors. And, please, check other figures in your paper regarding copyright.

Lines 521-522: “At pH range of 5.5 -521 12.5 lead ions exist as Pb (OH)°2 and as Pb (OH)°4 at pH greater than 12.5 [124].” Check Pb speciation at pH higher than 12.5.

General comment: Please, check your paper for mistype, punctuation, gap, spelling, etc.

Author Response

Reviewer #2

The manuscript Molecules-881517 is a well-organised review paper about multifunctional magnetic oxides nanoparticles (MNP’s) core-shell, that reports interesting aspects of MNP’s synthesis, structural studies and application for waste-water treatment. This paper can be slightly improved (please, see my comments below).

This paper is suitable for publication in Molecules Journal after minor revision.

Comments:

Lines 148-154: It would be good to add a table with brief description of each typical methods for the synthesis of magnetic nanoparticles and outlining advantages and disadvantages of these methods, maybe also include cost assessment.

Response

A table on the comparative preparation methods of magnetic nanoparticles using magnetite nanoparticles as an example has be included in the manuscript as advised.

Lines 347-350: Did you get permission from authors [10] to use their figure 1 in your paper? I think it is very important to get permission from the authors. And, please, check other figures in your paper regarding copyright.

Response

Author(s) permission whose figures or images were used in the manuscript have been well referenced and permission gotten were necessary.

Lines 521-522: “At pH range of 5.5 -521 12.5 lead ions exist as Pb (OH)°2 and as Pb (OH)°4 at pH greater than 12.5 [124].” Check Pb speciation at pH higher than 12.5.

Response

Proper formula for Pb(II) at pH 12.5 Pb(OH)42-  has been corrected in the manuscript.

General comment: Please, check your paper for mistype, punctuation, gap, spelling, etc.

Response

Manuscript has been carefully checked for grammatical and punctuation mistakes as advised.

Reviewer 3 Report

The authors describe here the use of magnetic nanomaterials and their relative hybrids for wastewater treatment. 

Line 102 and Figure 2 : I do not understand what functionality makes temperature responsive the system depicted on Figure 2.

I understand that it works at 30°C for Cu2+ and at 50°C for Pb2+. 

Table I: Ref 54: what cations ?

Table I: Ref 10 and 56: why cubic shape is not good for the application?

Generally speaking, there are many reviews on magnetic nanoparticles in the literature. 

In my opinion, is missing here information on what pollutants are targeted, where and in which quantity.

Then, new propects annd reflexion of future devices based on these systems are not discussed. The quantity of pollutants, the quantity of magnetic systems, the device and the operating conditions for an efficient depollution.

Moreover, the journal Molecules does not appear suitable for this paper. Materials or Nanomaterials seem to suit better.

Author Response

Reviewer #3

The authors describe here the use of magnetic nanomaterials and their relative hybrids for wastewater treatment. 

Line 102 and Figure 2: I do not understand what functionality makes temperature responsive the system depicted on Figure 2.

Response

The sentence has been expounded to communicate the scientific idea of the presentations. The band gap nanomaterial was link to their its solar responsiveness and properly referenced. 

I understand that it works at 30°C for Cu2+ and at 50°C for Pb2+. 

Table I: Ref 54: what cations?

Response

Pb2+, Ca2+, Cd2+ have been reflected in the communication.

Table I: Ref 10 and 56: why cubic shape is not good for the application?

Response

Comparatively, spherical shaped nanomaterials are desirable for wastewater treatment because of their relatively high kinetic energy and corresponding ease of clearance as compared to other shapes. This has been emphasised in the manuscript and referenced.  

Generally speaking, there are many reviews on magnetic nanoparticles in the literature.

Response

Reflected review papers in the manuscript have been limited to related discussions on magnetic nanomaterials as necessary to guide readers and avoid repetition of such scientific communications while focussing on the primary focus of the manuscript.

In my opinion, is missing here information on what pollutants are targeted, where and in which quantity.

Response

Table 3 gives an overview of the adsorbent mass and adsorbate volume used in the sorption process which was reflected as mass in (g/L). The type of pollutant(s) targeted and the adsorption capacity in (mg/g) which reflects the quantity of the adsorbate adsorbed at equilibrium are also captured in Table 3.

Then, new prospects and reflexion of future devices based on these systems are not discussed. The quantity of pollutants, the quantity of magnetic systems, the device and the operating conditions for an efficient depollution.

Response

Future prospect of magnetic nanomaterials and conditions for optimization for effective adsorption of pollutants and remediation of wastewater have been further expounded at the concluding portion of the manuscript as advised.

Moreover, the journal Molecules does not appear suitable for this paper. Materials or Nanomaterials seem to suit better.

Response

The manuscript is submitted to Molecules Materials Chemistry, advanced materials for environmental chemistry section. The manuscript is suit the special issue section to which it is submitted.

Reviewer 4 Report

This work discusses the possibility of using multifunctional magnetic oxides nanoparticles for waste-water treatment. Some critical note could direct to the authors.

  1. The authors claim that the mono-domain particles are superparamagnetic. They have to explain that more detailed. There is critical size for mono-domain state for each magnetic material and critical size for superparamagnetic state. Mono-domain particles might be superparamanteric at several conditions.
  2. The authors have to mention the advantages and disadvantages for the different methods of synthesis- particles size distribution, cost and ect.

Author Response

Reviewer #4

This work discusses the possibility of using multifunctional magnetic oxides nanoparticles for waste-water treatment. Some critical note could direct to the authors.

The authors claim that the mono-domain particles are superparamagnetic. They have to explain that more detailed. There is critical size for mono-domain state for each magnetic material and critical size for superparamagnetic state. Mono-domain particles might be superparamanteric at several conditions.

Response

Mono‑domain phase of magnetic nanomaterials and superparamagnetism have been symbiotically explained; also, the conditions for superparamagnetism were also highlighted as advised.

The authors have to mention the advantages and disadvantages for the different methods of synthesis- particles size distribution, cost and etc.

Response

A table on the comparative preparation methods for magnetic nanoparticles, advantages, disadvantages and cost using magnetite nanoparticles as example has be included in the manuscript as advised.

Round 2

Reviewer 3 Report

The paper has been improved.

Please, pay attention to Fig. 3 that is strongly distorded. 

Reviewer 4 Report

After the changes I recommend the manuscript to be published.